# Advancing Inflammatory Bowel Disease Treatment by Targeting the Innate Immune System and Precision Drug Delivery

**DOI:** 10.3390/ijms26020575

**Published:** 2025-01-11

**Authors:** Kat F. Kiilerich, Trine Andresen, Behrooz Darbani, Laura H. K. Gregersen, Anette Liljensøe, Tue B. Bennike, René Holm, Jesper B. Moeller, Vibeke Andersen

**Affiliations:** 1Department of Molecular Medicine, University of Southern Denmark, 5000 Odense, Denmark; kakiilerich@health.sdu.dk (K.F.K.); jbmoeller@health.sdu.dk (J.B.M.); 2Department of Health Science and Technology, The Faculty of Medicine, Aalborg University, 9220 Aalborg Ø, Denmark; trinea@hst.aau.dk (T.A.); tbe@hst.aau.dk (T.B.B.); 3Molecular Diagnostic and Clinical Research Unit, University Hospital of Southern Denmark, 6200 Aabenraa, Denmark; bds@rsyd.dk (B.D.); laura.gregersen@rsyd.dk (L.H.K.G.); a.liljensoe@rsyd.dk (A.L.); 4Department of Regional Health Research, University of Southern Denmark, 5000 Odense, Denmark; 5Department of Physics, Chemistry and Pharmacy, University of Southern Denmark, 5000 Odense, Denmark; reho@sdu.dk; 6Danish Institute for Advanced Study, University of Southern Denmark, 5000 Odense, Denmark

**Keywords:** inflammatory diseases, pharmacology, drug targets, optimize treatment, colonic drug delivery

## Abstract

Inflammatory bowel disease (IBD), encompassing Crohn’s disease and ulcerative colitis, involves chronic inflammation of the gastrointestinal tract. Current immune-modulating therapies are insufficient for 30–50% of patients or cause significant side effects, emphasizing the need for new treatments. Targeting the innate immune system and enhancing drug delivery to inflamed gut regions are promising strategies. Neutrophils play a central role in IBD by releasing reactive oxygen species (ROS) and neutrophil extracellular traps (NETs) —DNA-based structures with cytotoxic proteins—that contribute to mucosal damage and inflammation. Recent studies linking ROS production, DNA repair, and NET formation have identified NETs as potential therapeutic targets, with preclinical models showing positive outcomes from NET inhibition. Innovative oral drug delivery systems designed to target gut inflammation directly—without systemic absorption—could improve treatment precision and reduce side effects. Advanced formulations utilize properties such as particle size, surface modifications, and ROS-triggered release to selectively target the distal ileum and colon. A dual strategy that combines a deeper understanding of IBD pathophysiology to identify inflammation-related therapeutic targets with advanced drug delivery systems may offer significant promise. For instance, pairing NET inhibition with ROS-responsive nanocarriers could enhance treatment efficacy, though further research is needed. This synergistic approach has the potential to greatly improve outcomes for IBD patients.

## 1. Introduction

Inflammatory bowel disease (IBD) comprises a group of chronic conditions characterized by inflammation within the gastrointestinal system [1]. The main subgroups are ulcerative colitis (UC) and Crohn’s disease (CD) [2,3]. Typically emerging in early and middle adulthood, IBD can significantly interfere with education, work, and family life [4]. IBD affects more than 6.8 million people worldwide and is a growing concern due to the rising global incidence rates [5,6,7].

IBD is characterized by recurring flare-ups and the risk of permanent organ damage. Most treatments are focused on targeting immune-mediated inflammatory factors in adaptive immunity. However, despite the availability of efficient treatment options, 30–50% of patients experience insufficient response, severe side effects, or loss of treatment effect [8,9,10,11,12,13]. This underscores the urgent need for better treatment strategies.

Despite recent advances, the factors driving IBD remain incompletely understood [14,15,16]. Current research supports that IBD arises from a complex interplay of genetic predisposition and environmental factors, including diet [1,6]. It is further increasingly recognized that the intestinal microbiota, barrier function, and immune system play pivotal roles in IBD development, progression, and treatment outcomes [8,17].

While the specific mechanisms underlying the disease remain unclear, it is well established that the innate immune system is responsible for initiating inflammation in the gut. A compromised intestinal barrier, a fundamental component of the innate immune response, is critical to this process [1]. Neutrophils and neutrophil extracellular traps (NETs) not only capture and neutralize pathogens but also contribute significantly to this process by enhancing mucosal degradation and promoting barrier leakage [1,18]. The formation of NETs also contributes to the innate immune system’s capacity for memory through trained immunity. This emerging understanding of trained immunity highlights the ability of innate immune cells to adapt and optimize their responses to previously encountered challenges, indicating a more complex interplay between innate immune mechanisms and immunological memory than previously recognized [19]. This process subsequently helps activate the adaptive immune system, contributing to disease progression (Figure 1). Accordingly, targeting innate immunity may prove more effective in improving disease outcomes [20,21]. Therefore, focused research in this field, particularly in identifying and targeting IBD’s causal and early triggers, is of utmost importance in improving IBD treatment outcomes.

Parenteral, oral, and rectal routes are the classical methods of drug delivery, with oral administration being the most widely used. Oral pharmaceutical formulations aim to provide the systemic absorption of a compound. However, for IBD, which most commonly affects the intestines, this systemic approach often leads to limited therapeutic benefits and a high potential for adverse effects. Whereas conventional methods such as pH-sensitive coating are influenced by physiological changes in the gut of patients with IBD, leading to changes in pH and reduced drug effectiveness, newer methods, including ROS-dependent drug release nano-delivering systems, have shown promising effects in in vitro and animal studies [22,23,24]. Therefore, targeted drug delivery to gut inflammation sites could lead to better treatment of IBD.

In this review, we discuss promising future directions in IBD drug treatment, focusing on (i) targeting inflammatory processes in innate immunity and (ii) methods for targeted delivery to the inflamed intestine (Graphical abstract).

## 2. Search Strategy and Selection Criteria

Searches of PubMed and Web of Science identified data for this review up until 1 September 2024, using the following search terms alone or combined: Crohn’s disease, inflammatory bowel disease, ulcerative colitis, personalized medicine, pharmacology, drug delivery, innate immunity, NETs (neutrophil extracellular traps), targeted drug delivery, colonic drug delivery, drug delivery to the low intestine, and trained immunity. In addition, a backward citation search was completed on the included references. Only papers published in English were reviewed. The final reference list was generated based on originality and relevance to the broad scope of this review.

## 3. Current and Emerging Medical Treatment Options and Their Limitations

Several organizations, including The European Crohn’s and Colitis Organisation and the American Gastroenterological Association, have provided guidelines for treating adults with CD and UC [8,9,10,11,12].

Effective IBD treatment remains a significant challenge. Current therapies lack a cure, work for only 30–50% of patients, carry substantial side effects, and lack personalized approaches. Therefore, current medical management focuses on controlling inflammation and limiting disease progression by improving the efficacy of existing drugs and developing novel ones. The early initiation of biologic therapy or combination therapies might be beneficial for some patients, such as those with medically refractory IBD or extra-intestinal manifestations [25,26]. Infliximab combined with thiopurines exemplifies a successful strategy, demonstrating superior efficacy in achieving and maintaining remission compared to monotherapy [26].

Beyond current approaches, new avenues are emerging for UC and CD treatment, including new monoclonal antibody therapies, apheresis therapy, and the manipulation of the intestinal microbiome [27]. Novel drug targets include various pro-inflammatory components, such as αεβ7 integrins, Mucosal Vascular Addressin Cell Adhesion Molecule 1, Sphingosine-1 phosphate on T-cells, Interleukin-23 subunit p19, and Interleukin-12 subunit p40 [28,29].

Most of these approaches and emerging drugs have focused on the adaptive immune system. However, targeting components of the innate immune system may offer a more effective way to manage inflammation and achieve long-term remission [30]. This will be discussed in detail in the next section.

## 4. Targeting the Innate Immune System at the Gut Surface

The gut, diet, microbes, and immune factors interact closely. IBD is characterized by dysbiosis, which involves reduced microbiome diversity and disruption of gut microbiota homeostasis (Figure 1). The gut epithelial barrier serves as a crucial separation between the luminal contents and the underlying tissue layers, including immune cells. This epithelial barrier is compromised in IBD and explains why microbes can access and invade the mucosa, activate the innate immune system, and cause inflammation. Recent research has illuminated important roles of the innate immune system and trained immunity in IBD, highlighting neutrophils as a potential therapeutic target [31,32]. Neutrophils are mediators of inflammation through the recruitment and polarization of innate immune cells, such as pro-inflammatory M1 macrophages [33,34], while also regulating epithelial barrier integrity [31].

Mucosal degradation is a hallmark of IBD and is predominantly driven by the dysregulated infiltration of immune cells, such as neutrophils, to the gut. The neutrophil activation and subsequent release of reactive oxygen species (ROS) and proteases, including neutrophil elastase and collagenase, and matrix metalloproteinases, may lead to the destruction of the mucosa extracellular matrix, crypt distortion, and ulcerations [35,36].

In addition to ROS release, NETs have been reported in the gut mucosa during active inflammation in UC and CD [37,38,39]. Neutrophils generate NETs in a process known as NETosis in response to gut microbes such as bacteria and fungi [37]. NETs are web-like structures consisting of DNA, histones, and anti-microbial inflammatory proteins and proteases, including neutrophil elastase and myeloperoxidase, causing inflammation and damage to the surrounding tissue. Recent studies have linked NETs with inflammation [39,40,41] and found that NETs sustain inflammatory signals and mucosal degradation in UC [41] and may serve as a therapeutic target to reduce inflammation and mucosal degradation in IBD [40,42].

Novel NET-targeting therapeutics are currently being investigated in animal models of IBD [43]. Directly targeting NETosis by inhibiting protein arginine deiminases reduced colon inflammation in a dextrane sulphate sodium (DSS)-induced colitis mouse model [44,45]. Supporting this, Cyclosporine A can be applied as rescue therapy for acute, severe UC, and it has been demonstrated in a dextrane sulphate sodium-induced colitis mouse model that the anti-inflammatory effect of Cyclosporine A may be mediated by the inhibition of microbial lipopolysaccharide-stimulated NET release [46]. In addition, another study found that NET formation was significantly reduced in colons of UC patients who responded to treatment with Infliximab, a TNF inhibitor, further supporting the role of NETs as a potential target in gut inflammation [41].

Other strategies to reduce inflammation and tissue degradation are focused on enhancing NET degradation [47]. Although the in vivo mechanisms for clear NETs remain incompletely described, degradation by deoxyribonuclease I (DNase I) has been identified, and enhancing NET degradation through DNase I treatment has been demonstrated to alleviate colonic inflammation and decrease cytokine levels in several mouse models of IBD [40,48,49]. These findings suggest that inhibiting NET formation or abolishing NET structures could be explored as novel therapeutic targets in IBD and possibly other diseases.

Recent research highlights a complex interaction between ROS production, DNA repair, and NET formation, suggesting new therapeutic targets [50]. Since NETosis depends heavily on ROS production and subsequent DNA damage, modulating ROS levels or DNA repair could influence NETosis and associated inflammation [50]. Notably, hydrogen (H₂) appears to protect the intestinal barrier by inhibiting NET formation. In a rat model of hemorrhagic shock, MgH₂ reduced NET-related intestinal barrier damage by suppressing NET formation via a ROS/MAPK/PAD4-dependent pathway [51]. However, while DNA repair is critical in regulating NETosis, its specific role in gut inflammation remains unclear and warrants further study [52].

Moreover, with the development of immunotherapies, it is now possible to target cells and mechanisms of the innate immune system, as changes in the functional program of the innate immune system, through metabolic or epigenetic alterations, may lead to potential “therapeutic targets”. Consequently, suppressing epigenetic changes with histone or DNA methylation inhibitors using nanocarriers to deliver the drug to the desired immune cells might be a way forward [20,21]. However, to maximize the positive treatment effect and reduce adverse systemic effects, e.g., the systemic inhibition of the innate immune system, treatments should ideally be directed to the site of inflammation.

## 5. Drug Delivery to the Site of Inflammation

Parenteral, oral, and rectal routes are the classical methods of drug delivery, with oral administration being the most widely used. Oral pharmaceutical formulations are designed to achieve the systemic absorption of a compound. However, in the case of IBD, localized gut treatment may offer advantages over systemic therapies, assuming that an enhanced therapeutic benefit and a reduction in potential adverse effects are achieved. While rectal administration can be effective for patients with inflammation confined to the distal colon, it is less suitable when inflammation affects other regions of the intestine. Since IBD can impact any part of the gastrointestinal tract, there is a pressing need for the ability to target specific intestinal sites.

While targeting NETs has been demonstrated in experimental animal colitis models, targeting NETs in inflamed areas of the gut may theoretically enhance anti-inflammatory effects; however, documentation is still missing. Targeted drug delivery to inflamed areas could potentially improve the effectiveness of IBD treatment while minimizing systemic side effects. Several therapeutic delivery systems have been introduced, including nanoparticle-mediated drug delivery systems, enteric-coated microneedle pills, prodrug systems, lipid-based vesicular systems, hybrid drug delivery systems, and biologic drug delivery systems. This section provides an overview of the trends in potential drug delivery systems against IBD. For those interested, we refer to reviews that provide detailed information [53,54,55,56,57,58].

### Physiological Factors Relevant for Drug Delivery to IBD Patients

Although widely recognized, the significant physiological changes in the gut associated with chronic inflammation are often overlooked when developing pharmaceutical strategies for IBD patients. Inflammation-induced changes include (i) a compromised intestinal barrier due to ulcers, crypt distortions, and mucosal surface abnormalities, (ii) increased mucus production, and (iii) immune cell infiltration, including neutrophils, macrophages, lymphocytes, and dendritic cells. The result is that gut inflammation significantly affects drug delivery. Figure 2 describes the physiological factors relevant for drug delivery to IBD patients, such as gut microbes, altered transit time, and gut pH.

## 6. Conventional Pharmaceutical Approaches to Target the Distal Ileum and Colon

There are four classical pharmaceutical approaches for targeting the lower intestine [53,59].

pH-sensitive coatings: Tablets or pellets are coated with polymethacrylates, which solubilize at elevated pH levels, such as those found in the lower small intestine. This allows the drug to be released once the coating dissolves.Water-insoluble coatings: These involve the use of polysaccharides or wax, such as ethyl cellulose, which are insoluble in water. In incorporating soluble components into the coating, the release time can be controlled. Water slowly permeates the tablet, eventually causing the film to rupture and release the drug.Prodrug (microbially digested) coatings: Hydrophilic polysaccharides like pectin are used in coatings that are broken down by microbial activity in the distal parts of the gastrointestinal tract.Combination approach: These coatings can also be combined with other approaches to ensure drug release if one mechanism fails.

While these strategies are effective in healthy populations, they may be less successful in patients with IBD due to physiological changes associated with the disease, leading to inconsistent drug release [60,61]. For example, consider the following cases:In IBD patients, the pH may not increase sufficiently as the tablet moves through the gastrointestinal tract, preventing the solubilization of pH-sensitive coatings [60].Alterations in the gut microbiome may accelerate or reduce the digestion of coatings, potentially causing premature drug release in the upper intestine [61].Highly variable transit time makes it difficult to control time-released drug delivery and determine where in the intestine the drug is released.

A classic formulation available on the market is Entocort^®^ EC, a budesonide formulation designed for colon-targeted delivery in IBD. Entocort^®^ EC uses a pH-triggered, time-dependent release system. The granules are coated with Eudragit^®^ L, which dissolves at pH 5.5, ensuring budesonide release in the ileum and ascending colon. Additionally, ethyl cellulose in the formulation provides slow release in the colonic region [62]. However, despite this approach, classical pharmaceutical principles often fall short in delivering drugs effectively in IBD patients. This limitation is particularly critical for modern therapeutic modalities, such as peptides and siRNA, which may be released too early and be degraded by digestion enzymes before reaching their intended site of action [63].

Another conventional approach for colon-specific drug delivery is the prodrug strategy. A prodrug is a pharmacologically inactive compound that becomes active upon metabolism. This approach has been used to activate drugs in the colon. The most commonly described example is the azo conjugation of 5-aminosalicylate (5-ASA) [64,65]. 5-ASA is used for most patients with UC but demonstrates high individual variability in the pharmacological response due to upper gastrointestinal absorption and variation in gut microbial azoreductase activity [64,65]. Therefore, various methods have been applied to improve the delivery to the inflamed gut tissue [65,66,67]. One interesting approach is combining Mesalamine, a 5-ASA compound, with a mucoadhesive and cathepsin B-cleavable peptide [67]. This prodrug was demonstrated to accumulate within inflamed colons following oral administration in murine colitis models. Further, the prodrug was shown to enter intestinal macrophages and release Mesalamine after enzymatic cleavage [67].

Recently lipid-based gels with controlled lipolysis through surfactant selection have been suggested to ensure a late release in the intestinal tract; however, this has only been investigated in vitro [68]. Other chemical analogues have also been investigated as prodrugs [69,70]. However, the clinical value of these prodrugs has yet to be proven for the treatment of IBD. Altogether, there is a need for effective drug delivery strategies in IBD.

## 7. Novel Delivery Approaches

Given that classical pharmaceutical approaches may not be effective in IBD patients, various pharmaceutical formulation principles have been explored to improve drug efficacy. Although many of these principles have been known for years, they have yet to be adopted in clinical practice.

### 7.1. Nano-Delivery Systems

Nano-delivery systems (examples provided below) have gained attention as a means of drug delivery in IBD. These systems may be more effective at targeting the inflamed mucosa, providing better bioavailability at diseased tissue, reducing systemic adverse effects, and, finally, providing flexibility in delivery characteristics [71].

#### 7.1.1. Size-Based Targeting

The size of nano-delivery systems has been shown to improve colonic residence time in inflamed intestinal regions, thereby offering therapeutic benefits in IBD. Reduced particle size has been reported to enhance selective delivery to colitis-affected tissues through an enhanced epithelial permeability and retention effect [72,73,74,75,76,77,78]. This size-based approach also facilitates preferential uptake by immune cells, which is significantly increased in inflamed regions [75,79,80].

While smaller particles are often considered more effective for targeting infected tissues, contradictory reports exist. For instance, some studies have found that nano-sized delivery systems do not always exhibit specificity for diseased versus non-diseased tissues [81]. Generally, delivery specificity is reported to be particle size-dependent, with smaller diameters showing a more pronounced effect [75]. Particles studied have typically been manufactured using polymers such as polystyrene, poly-lactic-co-glycolic acid, polycaprolactone, and polyethylene glycol (PEG), where the active compound is encapsulated within the carrier system [79,81]. Based on published studies, the size-dependent effect appears relatively independent of the carrier material and is primarily driven by particle size.

Most of these studies are based on in vitro cell models, ex vivo tissue studies, or investigations on small animals. However, Schmidt and colleagues examined the uptake of nano- and microparticles in the rectal mucosa of human IBD patients [82]. They found an accumulation of microparticles in areas with active IBD but only trace amounts of nanoparticles, highlighting a potential lack of translatability from nonclinical to clinical conditions in IBD.

#### 7.1.2. Targeting Based on Surface Modifications

Modifying the surface properties of nanoparticles has been extensively investigated, including strategies such as PEGylation and the development of positively and negatively charged nano-delivery systems.

PEGylation: Adding polyethylene glycol (PEG) to nanoparticles creates a hydrophilic surface, which reduces interactions with the intestinal environment and enhances epithelial distribution [83,84,85,86]. Ex vivo studies suggest that the PEGylation of nanoparticles could be a viable pharmaceutical approach for ensuring accumulation in the inflamed colonic mucosa [78,83,87,88,89,90].Positively Charged Nanoparticles: Adding excipients, a substance added to the drug formulation, to create a positive surface charge on nanoparticles is theorized to increase adhesion to the mucosal surface, thereby enhancing the intestinal retention time of the delivery system [91]. This approach may also offer increased targeting potential in patients with CD, where increased mucosal production is often observed [92,93]. However, despite the theoretical benefits, studies exploring this strategy have not provided strong support for its effectiveness, suggesting that it may not be effective as a primary strategy [83,94,95,96,97,98,99,100].Negatively Charged Nanoparticles: Anionic delivery systems are designed to adhere to inflamed tissue through electrostatic interactions with the elevated concentration of proteins in inflamed regions [101,102,103]. Smaller particles are, in particular, reported to adhere more effectively to the mucus layer [73,75]. Studies investigating negatively charged nanoparticles for IBD treatment have generally shown positive results [83,90,104,105,106]. However, these particles are typically localized in the small intestine following oral administration [73,107,108]. Therefore, additional technologies would be required if the target is the colon.pH-sensitive hydrogels: Systems based on natural polymers such as alginate and chitosan have been proposed as carriers for IBD compounds or nano-emulsions to deliver low water-soluble compounds due to the polymers’ pH sensitivity and mucus adhesion. Positive results have been reported in nonclinical models; however, a key technical challenge could be the application of polymers derived from natural sources [109,110].

#### 7.1.3. Redox-Mediated Targeting

In patients with IBD, the increased production of ROS at the site of inflammation has been reported to result in a higher concentration of ROS in the mucosa of patients than in healthy subjects and to correlate with disease progression [111]. A targeted drug delivery system can exploit these physiological differences between patients and healthy subjects as the targeting mechanism. Redox-mediated drug delivery involves encapsulating the drug in polymeric nanoparticles designed to target inflammation in the colon. This method utilizes redox reactions with a high concentration of ROS. The nanoparticles consume the ROS while simultaneously releasing the drug at the site of inflammation, thereby enhancing therapeutic efficacy for treating colitis.

Consequently, redox-dependent nanoparticles have been investigated as a potential strategy for IBD treatment [24,112,113]. Studies have found that oxidation-sensitive materials [78,114], such as β-cyclodextrin [115,116,117], successfully released their cargo in response to ROS in the gastrointestinal tract. Shi et al. constructed an oral nano platform with dual enzyme/ROS sensitivity based on β-cyclodextrin and 4-(hydroxymethyl)phenylboronic acid [23]. The nano platform was used to carry Celastrol, a compound obtained from the Chinese plant Tripterygium wilfordii, and demonstrate its effectiveness in treating DSS-induced colitis in mice [23].

In another study, Wang et al. developed a carbohydrate-based bioresponsive nanosystem for IBD-specific drug delivery [22]. The multiple-carbohydrate-based nanosystem with pH/ROS dual responsibility and charge-mediated targeting ability was synthesized by modifying oxidation-sensitive cyclodextrin with 4-(hydroxymethyl) phenylboronic acid pinacol ester. Positively charged chitosan and negatively charged pectin were sequentially combined with cyclodextrin to form multiple polysaccharide-based nanosystems using the electrostatic self-assembly method. A nanosystem was loaded with dexamethasone and potently reduced inflammation in vitro in macrophages and in vivo in DSS-induced UC in mice [22].

While this approach appears promising for targeted drug delivery, its effectiveness may be limited by instability in low-pH, enzyme-rich environments and rapid drug release. Combining this approach with other technologies could enhance its potential for success; however, evidence in humans is, to the best of our knowledge, still absent.

#### 7.1.4. Active Targeting

Active targeting can be utilized to overcome physiological obstacles encountered during oral administration, such as enzymatic degradation. This can be achieved by coupling specific ligands, such as peptides or antibodies, to the surface of the nano-delivery systems. Several studies have investigated the potential of using targeting ligands for oral colon-specific drug delivery. Positive results have been reported for macrophage receptors, e.g., mannose receptors [118] and macrophage galactose-type lecithin [119], as well as transferrin receptors [120] and epithelial CD98 [121]. These studies have generally reported successful interactions between the target and the ligand in vitro and, to a lesser extent, in vivo. Thus, despite the potential of this approach, it remains technically complex. The technical challenges and high production costs associated with antibodies may restrict the widespread availability of this strategy.

### 7.2. Delivery Devices

The application of microneedles for intestinal drug delivery has been investigated in pigs. These microneedles are covered with a pH-sensitive coating that dissolves upon reaching the intestinal tissue, thereby exposing them. The peristaltic movements of the lumen ensure that the drug is released into the mucosa when the microneedles are near the mucosal tissue. Histological examination showed no tissue damage [122], suggesting that this concept may have the potential to deliver acid-labile drugs (e.g., siRNA). However, linking the delivery to a specific site in the intestines may be challenging in IBD patients due to highly variable intestinal transit times.

## 8. Conclusions: Challenges and Future Directions in IBD Drug Treatment

IBD remains a significant clinical challenge due to its complex pathology and variable patient response to existing therapies. While current treatments target adaptive immunity, recent evidence suggests that focusing on the innate immune system, particularly neutrophil extracellular traps (NETs), could provide more effective therapeutic outcomes. Additionally, advances in drug delivery systems, such as nanoparticle-based formulations and targeted delivery methods, offer the potential to improve treatment precision and reduce adverse effects by providing oral formulations able to target gut inflammation directly without systematic absorption. Therefore, a dual approach combining a better understanding of IBD pathophysiology to identify new therapeutic targets causally related to inflammation initiation with drug delivery systems that effectively target the inflamed intestine may be a promising approach. Hence, combining a promising drug target, such as NETs, with a promising drug delivery system using ROS that targets gut inflammation could be particularly interesting; however, currently, evidence supporting this is lacking. Such a strategy is crucial to achieving more specific and targeted treatment options for patients with IBD. Continued research into these novel approaches, including NET inhibition and site-specific drug delivery, is critical for developing more effective, personalized treatments for IBD.

## Figures and Tables

**Figure 1 ijms-26-00575-f001:**
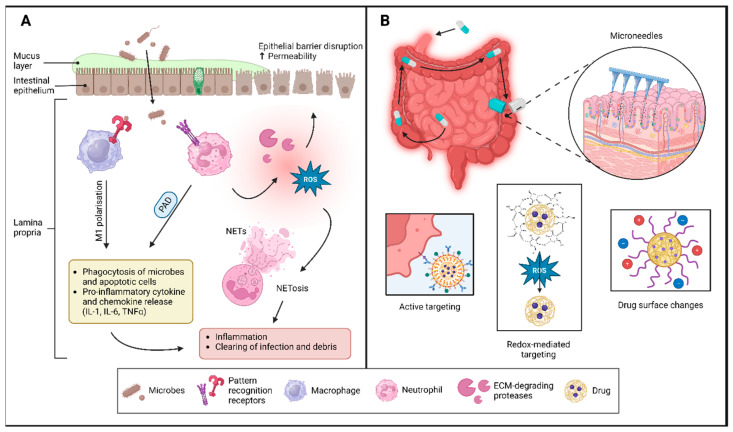
(**A**) Interactions between luminal content, such as gut-residing microbes, and the intestinal surface can activate innate immune responses that subsequently cause inflammation in the gut. The mechanisms with therapeutic potential are evaluated in this review. (**B**) Drug delivery strategies for targeting inflamed gut. Abbreviations: PAD, protein arginine deiminase; ROS, reactive oxygen species; NETs, neutrophil extracellular traps; ECM, extracellular matrix. Created in BioRender. Kiilerich, K.F. (2024) https://BioRender.com/h24q574 (accessed on 8 August 2024).

**Figure 2 ijms-26-00575-f002:**
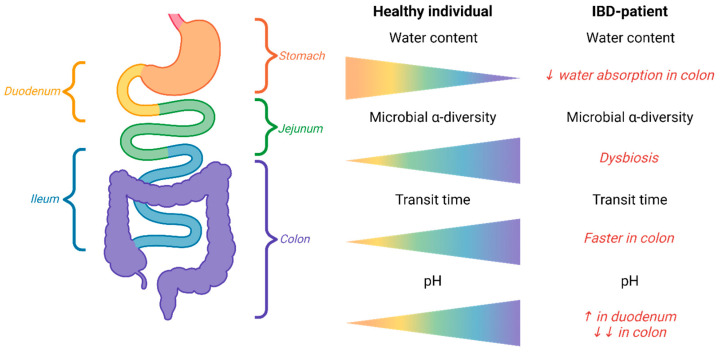
Physiological factors relevant for drug delivery to IBD patients. Physiological and microbial changes in IBD patients impact gastrointestinal motility, water resorption, pH, and microbiome diversity. Water content. In healthy individuals, water content decreases along the gastrointestinal tract, but in IBD patients, reduced water absorption in the colon leads to increased stool water content. Microbial diversity. Healthy individuals exhibit increasing microbial diversity through the gastrointestinal tract, whereas IBD patients show reduced diversity, known as dysbiosis. Transit time. In healthy individuals, drug passage time increases along the gastrointestinal tract. In IBD patients, the small intestine’s transit time is prolonged, while colonic transit is accelerated, complicating time- or site-specific drug delivery. pH. Healthy subjects experience a gradual pH decrease along the gastrointestinal tract. In IBD patients, pH is higher in the duodenum and lower in the colon compared to healthy individuals. Mucosal inflammation further disrupts motility, intestinal volume, and integrity, influencing both drug absorption and microbial metabolism. Created in BioRender. Kiilerich, K.F. (2024) BioRender.com/v02x038 (accessed on 16 August 2024), based on Hua et al., 2015 [54], with permission from Elsevier.

## Data Availability

No new data were created or analyzed in this study. Data sharing is not applicable to this article.

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
