# Peer review of "Advancing Inflammatory Bowel Disease Treatment by Targeting the Innate Immune System and Precision Drug Delivery"

_ijms, 2025, doi:10.3390/ijms26020575_

Round 1

Reviewer 1 Report

Comments and Suggestions for Authors

The study reviews inflammatory bowel disease treatment to target the innate immune system and precision drug delivery. The main question addressed by the research is the therapeutic strategies for inflammatory bowel disease. The topic is relevant to the field, and new treatments focused on the innate immune system is in great demand. This review describes drug delivery to the site of inflammation including redox-mediated targeting.

It is very important to investigate the inflammatory bowel disease and its treatment.

1. Figure 2 may be revised with additional explanation of the differences in the colon.

2. The underlines of some co-authors may be explained.

3. Since the production of ROS at the site of inflammation and oxidation-sensitive materials are very fundamental to treating inflammatory bowel disease, some additional discussion on the approaches may be added with some references in 7.1.3. Redox-mediated targeting.

Author Response

Reviewer 1

Comment 1

The study reviews inflammatory bowel disease treatment to target the innate immune system and precision drug delivery. The main question addressed by the research is the therapeutic strategies for inflammatory bowel disease. The topic is relevant to the field, and new treatments focused on the innate immune system is in great demand. This review describes drug delivery to the site of inflammation including redox-mediated targeting.

It is very important to investigate the inflammatory bowel disease and its treatment.

  1. Figure 2 may be revised with additional explanation of the differences in the colon.

Response 1

Thank you for pointing this out. We agree with this comment. Therefore, we have changes the description from:

IBD patients experience prolonged small intestine transit time and accelerated colonic transit, complicating drug delivery strategies reliant on timed or site-specific release. Colonic pH is significantly lower in IBD, affecting medications with pH-dependent coatings.

To:

225-232

Water content. In healthy individuals, water content decreases along the gastrointestinal tract, but in IBD patients, reduced water absorption in the colon leads to increased stool water content. Microbial diversity. Healthy individuals exhibit increasing microbial diversity through the gastrointestinal tract, whereas IBD patients show reduced diversity, known as dysbiosis. Transit time. In healthy individuals, drug passage time increases along the gastrointestinal tract. In IBD patients, small intestine transit time is prolonged, while colonic transit is accelerated, complicating time- or site-specific drug delivery. pH Healthy subjects experience a gradual pH decrease along the gastrointestinal tract. In IBD patients, pH is higher in the duodenum and lower in the colon compared to healthy individuals.

Comment 2

The underlines of some co-authors may be explained.

Response 2

Thank you for pointing this out. This was a mistake and has been deleted. We have marked the last authors and additionally changed the corresponding person as requested by email.

Comment 3

Since the production of ROS at the site of inflammation and oxidation-sensitive materials are very fundamental to treating inflammatory bowel disease, some additional discussion on the approaches may be added with some references in 7.1.3. Redox-mediated targeting.

Response 3

Thank you for pointing this out. We agree with this comment. Therefore, we have added some additional discussion and references (the added text in red):

364-391

7.1.3. Redox-mediated targeting

In patients with IBD, increased production of ROS at the site of the inflammation has been reported to result in a higher concentration of ROS in the mucosa of patients than in healthy subjects and to correlate with disease progression[1]. A targeted drug delivery system can exploit these physiological differences between patients and healthy subjects as the targeting mechanism. Redox-mediated drug delivery involves encapsulating the drug in polymeric nanoparticles designed to target inflammation in the colon. This method utilizes redox reactions with a high concentration of ROS. The nanoparticles consume the ROS while simultaneously releasing the drug at the site of inflammation, thereby enhancing therapeutic efficacy for treating colitis. Consequently, redox-dependent nanoparticles have been investigated as a potential strategy for IBD treatment[2-4]37288799. Studies found that oxidation-sensitive materials[5,6], such as β-cyclodextrin[7-9], successfully released their cargo in response to the ROS in the gastrointestinal tract. Shi et al. constructed an oral nano platform with dual enzyme/ROS sensitivity based on β-cyclodextrin and 4-(hydroxymethyl)phenylboronic acid[10]39061092. The nano platform was used to cargo Celastrol, a compound obtained from the Chinese plant Tripterygium wilfordii, and demonstrated its effectiveness in treating DSS-induced colitis in mice[10]39061092. In another study, Wang et al. developed a carbohydrate-based bioresponsive nanosystem for IBD-specific drug delivery39185345 [11]. The multiple carbohydrate-based nanosystem with pH/ROS dual responsibility and charge-mediated targeting ability was synthesized by modifying oxidation-sensitive cyclodextrin with 4-(hydroxymethyl) phenylboronic acid pinacol ester. Positively charged chitosan and negatively charged pectin were sequentially combined with the cyclodextrin to form multiple polysaccharide-based nanosystems using the electrostatic self-assembly method.  The nanosystem was loaded with dexamethasone and potently reduced inflammation in vitro in macrophages and in vivo in DSS-induced UC in mice39185345 [11]. While this approach appears promising for targeted drug delivery, its effectiveness may be limited by instability in low pH, enzyme-rich environments, and rapid drug release. Combining this approach with other technologies could enhance its potential for success, however, evidence in humans are to the best of our knowledge still absent.

Reviewer 2 Report

Comments and Suggestions for Authors

This manuscript focuses on therapeutic advances in IBD, specifically improving outcomes by targeting the innate immune system and precision drug delivery. The manuscript notes that IBD is a group of chronic diseases characterized by inflammation of the gastrointestinal tract, primarily including ulcerative colitis and Crohn's disease. Although existing treatments primarily target adaptive immunity, 30% to 50% of patients still have an inadequate response to therapy or experience severe side effects, making better treatment strategies urgently needed.

1. The article emphasizes the importance of the innate immune system in the pathogenesis of IBD, particularly the role of neutrophils and their external capture networks (NETs) in the initiation of inflammation. Therapeutic strategies targeting innate immunity may provide more effective treatment outcomes . But the role of precision drug delivery approaches is not described in detail.

2. The manuscript calls for more specific and targeted therapeutic options for patients with IBD by better understanding the pathophysiology of IBD, identifying new therapeutic targets associated with inflammation initiation, and incorporating effective drug delivery systems. But targeting those targets that may be more promising is not detailed.

 3. Why might localized intestinal therapies have a greater advantage over systemic therapies in the treatment of inflammatory bowel disease (IBD)? What are the specific mechanisms that may explain this advantage?  

4. How can nanomedicine delivery systems in the treatment of inflammatory bowel disease (IBD) enhance efficacy by improving drug localization in specific regions of the intestine? What research findings support this idea? 

 5. How do current therapeutic strategies for IBD balance individualized medicine with the choice of drug delivery systems to address the challenges of inadequate patient response to therapy and side effects?

Author Response

Reviewer 2

Comment 1

This manuscript focuses on therapeutic advances in IBD, specifically improving outcomes by targeting the innate immune system and precision drug delivery. The manuscript notes that IBD is a group of chronic diseases characterized by inflammation of the gastrointestinal tract, primarily including ulcerative colitis and Crohn's disease. Although existing treatments primarily target adaptive immunity, 30% to 50% of patients still have an inadequate response to therapy or experience severe side effects, making better treatment strategies urgently needed.

  1. The article emphasizes the importance of the innate immune system in the pathogenesis of IBD, particularly the role of neutrophils and their external capture networks (NETs) in the initiation of inflammation. Therapeutic strategies targeting innate immunity may provide more effective treatment outcomes . But the role of precision drug delivery approaches is not described in detail.

Response 1

Thank you for pointing this out. We agree with this comment. The role of precision drug delivery approaches is described in 5. Drug delivery to the site of inflammation and the following sections 6. Conventional pharmaceutical approaches to target the distal ileum and colon and 7. Novel delivery approaches. 

We have added the following to the introduction (marked in red)

74-78

Parenteral, oral, and rectal routes are the classical methods of drug delivery, with oral administration being the most widely used. Oral pharmaceutical formulations aim to provide systemic absorption of a compound. However, for IBD, which most commonly affects the intestines, this systemic approach often leads to limited therapeutic benefits and a high potential for adverse effects.  Whereas conventional methods such as pH-sensitive coating are influenced by physiological changes in the gut of patients with IBD, leading to changes in pH and reduced drug effectiveness, newer methods, including ROS-dependent drug release nano-delivering systems, have shown promising effects in in vitro and animal studies. Therefore, targeted drug delivery to gut inflammation sites could lead to better treatment of IBD.

Comment 2

The manuscript calls for more specific and targeted therapeutic options for patients with IBD by better understanding the pathophysiology of IBD, identifying new therapeutic targets associated with inflammation initiation, and incorporating effective drug delivery systems. But targeting those targets that may be more promising is not detailed.

Response 2

Thank you for pointing this out. We agree with this comment. Therefore, we have highlighted the link between NETs and ROS:

To section 4. Targeting the innate immune system:

165-172

Recent research highlights a complex interaction between ROS production, DNA repair, and NET formation, suggesting new therapeutic targets. Since NETosis depends heavily on ROS production and subsequent DNA damage, modulating ROS levels or DNA repair could influence NETosis and associated inflammation. Notably, hydrogen (H₂) appears to protect the intestinal barrier by inhibiting NET formation. In a rat model of hemorrhagic shock, MgH₂ reduced NET-related intestinal barrier damage by suppressing NET formation via a ROS/MAPK/PAD4-dependent pathway. However, while DNA repair is critical in regulating NETosis, its specific role in gut inflammation remains unclear and warrants further study.

To section  5. Drug delivery to the site of inflammation:

199-201

While targeting NETs has been demonstrated in experimental animal colitis models, targeting NETs in inflamed areas of the gut may theoretically enhance anti-inflammatory effects, however, documentation is still missing.

And to conclusion:

421-423

IBD remains a significant clinical challenge due to its complex pathology and variable patient response to existing therapies. While current treatments target adaptive immunity, recent evidence suggests that focusing on the innate immune system, particularly neutrophil extracellular traps (NETs), could provide more effective therapeutic outcomes. Additionally, advances in drug delivery systems, such as nanoparticle-based formulations and targeted delivery methods, offer the potential to improve treatment precision and reduce adverse effects. Therefore, a dual approach by combining a better understanding of IBD pathophysiology to identify new therapeutic targets causally related to inflammation initiation with drug delivery systems that effectively target the inflamed intestine may be a promising approach. Hence, combining a promising drug target, NETs, with a promising drug delivery system using ROS targeting gut inflammation could be particularly interesting; however, currently, evidence supporting this is lacking. Such a strategy is crucial to achieving more specific and targeted treatment options for patients with IBD. Continued research into these novel approaches, including NET inhibition and site-specific drug delivery, is critical for developing more effective, personalized treatments for IBD.

Comment 3

Why might localized intestinal therapies have a greater advantage over systemic therapies in the treatment of inflammatory bowel disease (IBD)? What are the specific mechanisms that may explain this advantage?  

Response 3

Thank you for pointing this out. We agree with this comment.

Therefore, we have added to the discussion

417-418

IBD remains a significant clinical challenge due to its complex pathology and variable patient response to existing therapies. While current treatments target adaptive immunity, recent evidence suggests that focusing on the innate immune system, particularly neutrophil extracellular traps (NETs), could provide more effective therapeutic outcomes. Additionally, advances in drug delivery systems, such as nanoparticle-based formulations and targeted delivery methods, offer the potential to improve treatment precision and reduce adverse effects by providing oral formulations able to target gut inflammation directly without systematic absorption. Therefore, a dual approach by combining a better understanding of IBD pathophysiology to identify new therapeutic targets causally related to inflammation initiation with drug delivery systems that effectively target the inflamed intestine may be a promising approach. In other words, Such a strategy is crucial to achieving more specific and targeted treatment options for patients with IBD. Continued research into these novel approaches, including NET inhibition and site-specific drug delivery, is critical for developing more effective, personalized treatments for IBD.

Comment 4

How can nanomedicine delivery systems in the treatment of inflammatory bowel disease (IBD) enhance efficacy by improving drug localization in specific regions of the intestine? What research findings support this idea? 

Response 4

Thank you for pointing this out. The hypothesis for the intestinal targeted delivery is that a higher concentration of the active compound will be available at the site where the disease is. Hence, the chance for a desired pharmacological response is higher, while the unwanted effects remain as low as possible. The reviewer is completely right in the comment, there is no research finding that specifically have proven that the strategy is beneficial, but it’s rather a general assumption in the field as also defined in the way we have described it in section 5, however, to clarify this point, we have modified the section, where the element of local delivery is mentioned.

  1. Drug delivery to the site of inflammation

193-194

Parenteral, oral, and rectal routes are the classical methods of drug delivery, with oral administration being the most widely used. Oral pharmaceutical formulations are designed to achieve systemic absorption of a compound. However, in the case of IBD, localized gut treatment may offer advantages over systemic therapies, assuming that an enhanced therapeutic benefit and a reduction of potential adverse effects are achieved.  

Comment 5

How do current therapeutic strategies for IBD balance individualized medicine with the choice of drug delivery systems to address the challenges of inadequate patient response to therapy and side effects?

Response 5

Thank you for pointing this out. We agree with this comment. Therefore, we have changed:

Since a cure for IBD is not available, current medical management focuses on controlling inflammation and limiting disease progression by improving the efficacy of existing drugs and developing novel ones.

To:

100-102

Effective IBD treatment remains a significant challenge. Current therapies lack a cure, work for only 30-50% of patients, carry substantial side effects, and lack personalized approaches.